# Microstructural Evolution and Failure in Fibrous Network Materials: Failure Mode Transition from the Competition between Bond and Fiber

**DOI:** 10.3390/ma17092110

**Published:** 2024-04-29

**Authors:** Yao Zhang, Weihua Wang, Pengfei Wang, Zixing Lu, Zhenyu Yang

**Affiliations:** 1Advanced Materials and Energy Center, China Academy of Aerospace Science and Innovation, Beijing 100088, China; weihuawang2011@163.com (W.W.); hvhe@163.com (P.W.); 2Institute of Solid Mechanics, Beihang University (BUAA), Beijing 100083, China; luzixing@buaa.edu.cn (Z.L.); zyyang@buaa.edu.cn (Z.Y.); 3Aircraft & Propulsion Laboratory, Ningbo Institute of Technology (NIT), Beihang University (BUAA), Ningbo 315832, China

**Keywords:** fibrous network materials, damage evolution, failure modes transition, finite element method (FEM), carbon, ceramics

## Abstract

For the complex structure of fibrous network materials, it is a challenge to analyze the network strength and deformation mechanism. Here, we identify a failure mode transition within the network material comprising brittle fibers and bonds, which is related to the strength ratio of the bond to the fiber. A failure criterion for this type of fibrous network is proposed to quantitatively characterize this transition between bond damage and fiber damage. Additionally, tensile experiments on carbon and ceramic fibrous network materials were conducted, and the experimental results show that the failure modes of these network materials satisfy the theoretical prediction. The relationship between the failure mode, the relative density of network and strength of the components is established based on finite element analysis of the 3D network model. The failure mode transforms from bond damage to fiber damage as increasing of bond strength. According to the transition of the failure modes in the brittle fibrous network, it is possible to tailor the mechanical properties of fibrous network material by balancing the competition between bond and fiber properties, which is significant for optimizing material design and engineering applications.

## 1. Introduction

Fibrous network materials consist of stochastic and interconnected fibers, and are widely found in both biological [1] and man-made materials [2]. Due to their stochastic structure, fibrous networks are heterogeneous, and their deformation is non-affine, posing a significant challenge in characterizing their mechanical properties. Experimental studies on fibrous networks, such as non-woven composites [3,4], paper [5], insulation tiles [6,7,8,9], sintered metal fibers [10,11], and biological tissues [1,12], have qualitatively established the relationship between density, network anisotropy, and network strength. Several numerical models have been proposed to investigate the intrinsic deformation mechanism of fibrous networks. In some network models obtained by computed tomography images, fibers were generally modeled using beam elements, truss elements, or solid elements [13,14], while the bonds between fibers were assumed to be nonlinear springs [15,16,17], rigid connections [18,19], trusses [20], beams [6,7,21,22,23], connector constraints [24,25], or “stick-slip” nonlinear contacts [26].

The mechanism of fibrous materials is distinctive from fiber-reinforced composite [27,28]. Based on the numerical simulations, the influence of the fibrous structure and properties of fibers and bonds on the mechanical properties of the network was revealed. Stergios et al. [20] investigated the mechanical properties of celluloses, and they found that bond density and bond strength enhance the elastic modulus and strength of the fibrous network materials, yet have a minor effect on the failure strain. Borodulina et al. [29] concluded that increasing the average number of bonds on a fiber could improve the mechanical properties of the network. Negi et al. [30] studied the influence of inter-fiber adhesion on the mechanical behavior of 2D fibrous networks, and showed that an increase in the adhesion strength between fibers reduces the network’s elastic modulus. A comparison between the random fibrous network and the cellular structure was also discussed [31], which demonstrated that the network’s strength was independent of the fiber tortuosity and fiber properties, while the network strength was related to the bond strength and the length of fiber segments.

Generally, the network strength depends not only on network structure, such as bond density (or average bond number per fiber) and the average length of fiber segments, but also on the properties of network components, such as bond strength and fiber strength. Deogekar and Picu [32] focused on the failure modes and the relationship between network structural parameters and network strength, and they demonstrated that failure of the fibrous network was related to the breakage of bonds. Malakhovesky et al. [33] found that, for the network for which the main failure mode was fiber damage, enhancing the variability of fiber strength could reduce the network strength. On the other hand, for the failure of networks associated with bond failures, the strength distribution among the bonds shows no significant influence on the elastic modulus and tensile strength of fibrous network [29]. It was shown that failure modes such as fiber damage or bond damage in the fibrous network are essential for enhancing the network strength. Additionally, Estelle et al. [34] studied the influence of heterogeneities on the failure behavior of disordered 2D lattices, and found that tuning the connectivity of the network could change the failure mode of network from brittle failure to ductile failure. Luo et al. [35] built a 3D transverse fiber network model, and studied the influence of the anisotropy on its mechanical properties. The numerical results indicated a brittle-to-ductile transition in the fibrous network with the anisotropy of network increasing. Thus, whether there existed a failure mode transition between fiber damage and bond damage in the fibrous network needs quantitative description.

In this work, the transition of failure modes in fibrous network materials is analyzed by considering the strength ratio of bond to fiber, and the failure mechanisms of these materials are discussed in detail. The influence of the network’s relative density and the mechanical properties of fibers and bonds on network strength is studied numerically using a 3D fibrous network model [6,7]. Through investigating damage evolution and stress distribution in the network, a failure criterion is established to quantitatively identify the failure modes of the fibrous network material. Additionally, uniaxial tensile tests on carbon and ceramic fibrous network materials are conducted, and the experimental results are in agreement with the theoretical predictions.

## 2. Material and Method

### 2.1. Experimental Section

#### 2.1.1. Materials

Two types of fibrous network materials are selected to examine the failure modes, including carbon fibrous network material and ceramic fibrous network material. The carbon fibrous network materials consist of phenolic resin and chopped rayon-based carbon fibers, with a relative density of 0.14. The ceramic fibrous materials contain silica fibers and mullite fibers, with a relative density of 0.16. These fibers are bound by a sintering additive (B4C powder and soluble starch) after undergoing the sintering process. The manufacturing processes for the carbon and ceramic fibrous materials are illustrated in Figure 1.

Chopped carbon fibers (30 wt.%) with an average length of 0.8 mm and an average diameter of 9 μm are dispersed in water. A dispersant (1.5 wt.%), such as polyacrylamide, is added to the solution to prevent fiber aggregation in the slurry. After sufficient high-speed stirring, the chopped fiber slurry is formed. Subsequently, a phenolic resin solution (60 wt.%) diluted with ethanol is added to the slurry. Following vigorous stirring for over an hour, a thoroughly mixed slurry is obtained.

Then, the slurry described above was poured into a mold. At the bottom of the mold, a plate with a uniform distribution of 1 mm diameter holes was covered with porous gauze (200 meshes with a mesh size of 76 μm). The excess solution was extruded from the holes under vacuum pressure (approximately 0.1 MPa). After sufficient drying, the carbon fibrous preform was sintered in a furnace and heated to 1200 °C in an argon atmosphere at a rate of 5 °C/min. During the sintering process, the chopped carbon fibers were bonded together by the carbonization of the phenolic resin. After cooling down to room temperature within the furnace, the carbon fibrous network materials were completed.

The fabrication process of the ceramic fibrous network is similar to that of the carbon fibrous network. Silica fibers and mullite fibers are used in the ceramic fibrous network materials. These two types of fibers, comprising about 70% silica fibers and 30% mullite fibers by weight, are first cut into short fibers with an average length of 0.8 mm and an average diameter of 10 μm. Then, these short fibers are mixed into the slurry along with the sintering additive (B_4_C powder and soluble starch). The sintering temperature is set at 1200 °C in an air atmosphere. During the sintering process, the B_4_C powder is oxidized at high temperatures, and the random fibers are bonded together by the B_2_O_3_ bond materials, resulting in the formation of the ceramic fibrous network materials.

#### 2.1.2. Quasi-Static Tensile Tests

Fibrous network material is a novel lightweight porous material, and there is currently no universal testing standard. For reference, we have reviewed some of the relevant research works [34,35,36,37,38,39,40,41,42,43,44,45,46,47,48,49]. Highly porous materials are sensitive to their microstructure; therefore, we believe that using appropriately large specimen dimensions can help reduce the dispersion of experimental data. The experimental specimens are designed with a dumbbell shape and a square cross-section measuring 75 mm × 15 mm × 15 mm, as illustrated in Figure 2a. The quasi-static uniaxial tensile tests on the specimens are performed using an in situ biaxial mechanical testing machine, IPFB-2000, at a loading rate of 0.2 mm/s. An electron scanning microscope (SEM, Model S-570, Hitachi, Tokyo, Japan) is used to observe the fracture surface after the tensile test to identify the typical failure modes of the fibrous network material.

### 2.2. Numerical Simulations

#### 2.2.1. Fibrous Network Model

The 3D fibrous network models are established based on the experimental observation of network materials. A random fiber in a 3D domain can be characterized by the coordinates of its mass point, *M_i_*, and two Euler angles, *α* and *β*, as shown in Figure 3d. Based on the porosity and domain size, fibers are generated randomly in the domain, and any two fibers are connected with each other through the bond when their least distance is within the bonding distance, as shown in Figure 3c. In this model, these fibers across boundary of the domain are cut into two pieces, and the portion outside the boundary is shifted to the opposite boundary. Therefore, periodic boundary condition could be applied to this 3D geometrical model of fibrous network. Finally, the representative volume element (RVE) of the random fibrous network model can be established (Figure 3b). The detailed establishment of the 3D fibrous network model can be found in previous studies [6,7].

#### 2.2.2. Finite Element Analysis

The quasi-static tensile test is simulated using ABAQUS 2016/Explicit software. Considering the aspect ratio of fiber is below 10, a Timoshenko beam is used to simulate the fibers in the complex network, for computational efficiency. The B31 element type of the Timoshenko beam in ABAQUS 2016 is taken in the following FEM simulations. There are many mechanical models proposed for the simulation of bonding material, whose function is to constrain deformation of fibers and transmit the load between fibers. Based on the previous numerical investigation on ceramic fibrous network [6,7,8,9,21,23], the bond material transmits the moments and forces before its breakage and can be reasonably modeled as Timoshenko beam element with a circular cross-section. In order to reflect the interaction in the fibrous network, fiber-to-fiber contact is incorporated into the simulation and defined as the general contact interactions containing hard contact in the normal direction and sliding friction in the tangential direction. The constitutive relationships of fiber and bond are both elastic until their failure, which is reasonable for the fibrous network fabricated by brittle fibers, such as carbon fibers and ceramic fibers. The maximum principal stress criterion is adopted via the subroutine of user define material (VUMAT) in ABAQUS 2020 software. The 3D numerical model is subjected to the uniaxial loading along the *z*-axis, as shown in Figure 3a.

## 3. Results

### 3.1. Experimental Results

The tensile stress–strain curves for the carbon fibrous network and the ceramic fibrous network are illustrated in Figure 4. For both networks, the stress–strain curves are similar to those of typically brittle materials. The stress–strain curve of the ceramic network material is linear up to a strain of 0.5%. Then, the stress increases nonlinearly with the strain, and there is a sudden drop in stress at a strain of 0.7% due to progressive damage in the fibrous network. For the carbon network material, the slope of the linear portion of the curve is steeper than that of the ceramic network materials, indicating a higher elastic modulus, with the fracture strain measured at 0.14%. This suggests that the elastic modulus of the carbon network material is larger than that of the ceramic network material. However, the tensile strength of the carbon network is lower than that of the ceramic network material. After the tensile tests, the fracture surfaces of the ceramic and carbon networks were observed with SEM. In Figure 4b, the bonds appear nearly intact, while most fibers are broken near the bonds, indicating that fiber damage is the predominant failure mode for the ceramic network material. In contrast, Figure 4c shows that for the carbon fibrous network, the fibers remain mostly intact, and the bonds exhibit breakage, attributable to the weak mechanical properties of the pyrolytic carbon bonds. This reveals that the primary failure mode for the carbon network material is bond damage.

### 3.2. Numerical Results

Both the carbon fibrous network and the ceramic fibrous network are composed of brittle fibers and brittle bonds, but the relative strength of fibers to bonds differs significantly. Based on the experimental results, the failure mode of fibrous network materials is clearly distinct, attributable to the different mechanical properties of the fibers and bonds. We have established 3D numerical models that replicate the geometric parameters and material properties (listed in Table 1) of these fibrous network materials. To elucidate the failure mechanism and understand the failure modes within the fibrous network, a failure criterion for network materials with brittle fibers and bonds is proposed as follows:
(1)$$R=\frac{S_{b}/\sigma_{b}}{S_{f}/\sigma_{f}}\;\{\begin{array}{c} R<1,\;\mathrm{fiber damage}\\ R>1,\;\mathrm{bond damage} \end{array}$$
where *σ*_b_ and *σ*_f_ represent the strength of the bond and the fiber, respectively; *S*_b_ and *S*_f_ are the current principal stress in the bonds and fibers, respectively. *R* = 1 corresponds to the critical failure transition of the fibrous network. The failure mode tends to be characterized as fiber damage dominated for *R* < 1, and tends to be bond damage dominated for *R* > 1.

Quasi-static tensile simulations were performed with finite element software ABAQUS/Explicit software 2016. The 3D fibrous network model was simulated by Timoshenko beam elements (B31 in ABAQUS 2016). A fiber was meshed into 60~80 B31 elements according to the length of the fiber. The number of elements is based on the relative density of the fibrous network model. For instance, models with a relative density of 0.08 have an average of 12,552 elements, including 4616 elements for bonds and 7936 elements for fibers. Conversely, models with a relative density of 0.45 contain an average of 172,340 elements, including 101,203 for bonds and 71,137 for fibers. The boundary conditions were periodic boundary conditions, chosen to closely match those of the uniaxial tensile experiments. The calculation error of numerical results due to the analysis method is controlled by the stability limit Δ*t* = (*L_e_*)_min_/*f*c, where (*L_e_*)_min_ is the minimum axial length among all the beam elements in the network, and $c=\sqrt{E/\rho }$ is the wave speed of the materials, which depends on the elastic modulus *E* and the density *ρ*. The mass scaling parameter *f* is used to improve computational efficiency. In this section, we discuss the friction coefficient, mass scale, and loading strain ratio in order to obtain stable parameters based on the numerical model with porosity = 94% (comprising 9170 beam elements). Figure 5 shows the out-of-plane compressive stress–strain curves at different mass scales (Figure 5a), friction coefficients (Figure 5b), loading strain ratios (Figure 5c). Here, friction coefficients of 0, 0.25, 0.5, and 1.0 were used to simulate fiber-fiber sliding behavior, which has also been employed in other simulation studies related to the sliding behavior of fiber networks [16] and foam-like monolithic carbon materials [50].

As shown in Figure 5a, the mass scale can enhance the computation and also increase the inertia effect, which reflects on the increasing the fluctuation of stress–strain curves, especially for the pseudo-elasticity part. Then, we discussed the friction coefficient at mass scale f = 4900. Friction can increase the loading transfer between fibers. If we focus on a fiber of the fiber network, and the others can be treated as matrix, then increasing the friction will be equivalent to enhancing the interfacial strength between fiber and matrix. With friction coefficient increasing, the stress–strain curves tend to be stable after friction coefficient = 0.5. Finally, we determine a set of parameters: Mass scale is 900, friction coefficient is 0.5, and loading strain ratio is 1.0 s^−1^, in order to obtain the balance between the computational efficiency and the stable numerical results.

In the numerical simulations, the principal stress *S*_11_ is obtained at the centroid integration point in the B31 element when a uniaxial strain of 0.1% is applied to the network model. *S*_b_ and *S*_f_ are the average of principal axial stress *S*_11_ in the bond elements and the fiber elements throughout the entire network model, as shown in Figure 6a,b. This failure criterion is verified through tensile experiments on two types of fibrous network materials. Based on SEM observations of the specimens, the predominant failure modes for the carbon fibrous network and the ceramic fibrous network are bond damage and fiber damage, respectively, as illustrated in Figure 4b,c. According to the failure criterion, the value of *R* for these two fibrous network materials is 5.26 and 1.01, respectively. The experimental results are in good agreement with the predictions made by the failure criterion.

## 4. Discussion

In order to study the effects of the mechanical properties of component materials on the failure mode of fibrous network materials, we define a ratio of bond strength *σ*_b_ to fiber strength *σ*_f_ as *K* = *σ*_b_/*σ*_f_. The following simulations are not limited to one specific material system. The detailed geometric parameters and material properties for the numerical model are listed in Table 2. In these simulations, the strength of the fiber is held constant at 1.0 GPa, while the bond strength varies from 0.1 GPa to 10 GPa.

Figure 7a illustrates the tensile stress–strain curves of the network models with *K* ranging from 0.1 to 10. In the elastic regime, the slopes of all the stress–strain curves are almost the same. The maximum tensile stress in the stress–strain curves is defined as the tensile strength of fibrous network *σ*_t,_ with the *K* increasing from 0.1 to 4.0, *σ*_t_ increases from 0.07 MPa into 1.6 MPa. When *K* > 4.0, it is obvious that *σ*_t_ almost maintains 1.6 MPa with the increasing of *K*. After the maximum tensile stress is reached, the overall stress begins to decrease with the strain due to the accumulation of damage in the network. The strain corresponding to the tensile strength of the network is defined as failure strain *ε*_f_. Then, the statistics of damaged bonds and damaged fibers in the networks can be obtained at the failure strains in all the cases. As shown in Figure 7b, the difference between the damage evolution of bonds and fibers are distinct. With the continuous increase in *K*, the percentage of damaged fibers increases. After *K* reaches to 4.0, the percentage of damaged fibers no longer rises, and remains an almost constant value of 1.5%. On the other hand, the percentage of the damaged bonds increases linearly and reaches the maximum at *K* = 1.0 with the increase in *K*. Then, with the continuous increase in *K*, the percentage of damaged bonds decreases. When *K* is larger than 4.0, the percentage of damaged bonds no longer increases and remains stable (below 0.2%). In other works by Stergios [20] and Borodulina [29], it is shown that the mechanical properties of the fibrous network increase with the increasing of bond strength. In their network model, the corresponding values of *K* increased from 0.01 to 0.1, which agrees with our simulation results. In addition, based on the damage evolution in the network, the percentage of damaged fibers and bonds no longer changed after *K* increases to 4.0, which further causes the tensile strength and failure strain of the network also remain constant. It is found that the influence of bond strength for increasing the network strength has an upper bound.

According to the numerical results, the distribution of the damage in fibers and bonds in the network is plotted in Figure 8. The value *R* of each case with different *K* is also calculated and marked near the double arrow. It is demonstrated that the *R* of case with *K* = 2 is 1.05 which is close to the critical failure transition. With increasing of *K*, the value of *R* decreases and the failure mode tend to be the bond damage dominated.

In the case of *K* = 0.5, plenty of bond damage is distributed randomly in the network at the strain of 0.01, and only a few fibers damage near the loaded end, which may be attributed to the constrain at the boundary. With the increasing of loading, the amount of damaged bonds increases rapidly. In the case with *K* = 1, the amount of damaged bonds at the strain of 0.01 is obviously less than that in the case with *K* = 0.5. With the increase in the bond strength, the amount of damaged bonds declines, while nearly no damaged bonds can be observed in the fibrous network at the strain of 0.01 with *K* = 5 or *K* = 10. Therefore, it is obvious that a failure mode transition between the fiber damage and the bond exists in the networks with the value of *K* ranging from 0.1 to 10. The numerical results also satisfied the prediction of the failure criterion.

In addition, the influence of relative density on the failure mechanism is also discussed. The numerical models of the fibrous network with the relative density ranging from 0.08 to 0.45 are generated. In these models, the ratio of bond strength and fiber strength, *K*, ranges from 0.1 to 10. The relationship among the variable *R*, the relative density and variable *K* is depicted by a curved surface, *R*-surface in Figure 9. The *R*-surface intersects with the surface of *R* = 1, the critical failure surface. The region of the *R*-surface colored with red above the critical failure surface means that the failure mode of network tends to be the bond damage. As the value of *R* increases, the bond damage tends to be more obvious. The region of *R*-surface colored with blue below the critical failure surface means the failure mode of network tends to be the fiber damage. As the value of *R* decreases, the fiber damage tends to be more obvious.

In most non-woven network materials, the bond damage is a dominated damage mechanism in the fibrous networks [49,50,51,52]. Based on the mechanical properties of non-woven materials determined by Chen [53], the relative density of non-woven SF20 and the value of *K* are about 0.2 and 0.4, respectively. According to the deformation mechanism of the network, the load in fiber can be released by rotation or bending of the fibers. While the deformation of the bond is restrained by the connected fibers, which leads to the stress in bond being higher than that in fiber in the network with small deformation, indicating *S*_b_:*S*_f_ > 1. Thus, the value of *R* for SF20 is at least 2.5. The failure mode of the non-woven network is characterized as bond damage, which agrees well with our prediction by the proposed failure criterion.

Furthermore, the other simulations of paper network [24] show that the network strength is mainly controlled by the inter-fiber bonds. In these works, the relative density of the paper network is 0.6~0.8, and the variable *K* is 0.01. Based on our failure criterion, the value of *R* ranges from 5 to 10 in the network model with *K* = 0.1. For the paper network with *K* = 0.01, we believe the value of *R* will be higher than 1, even if the relative density is 0.6~0.8. The failure mode of paper network is the bond damage domain, which is because the strength of paper network highly depends on the bond strength.

## 5. Conclusions

In this paper, the typical failure modes of the fibrous network materials are analyzed by combing the numerical simulations and experiments. A failure criterion is proposed especially for the network materials with brittle fibers and bonds, to quantitatively characterize the failure mode transition between bond damage and fiber damage, which is then verified by the tensile experimental results of two kinds of fibrous network materials. The failure mode of other network materials, such as paper and non-woven materials, is also coincidentally well within the theoretical prediction. The detailed conclusions are as follows:Based on the numerical results of 3D network models, it was found that the failure mode transition of network is from the bond damage domain to the fiber damage domain.For the variable *R* in the failure criterion, *R* > 1 means the failure mode of network tends to be bond damage; *R* < 1 indicates the failure mode of network tends to be fiber damage. This formula is verified by our experimental results, and is to be further confirmed by other network materials, such as paper and non-woven network.The failure mode of network depends on the network structure and mechanical properties of fibers and bonds. The value of variable *R* decreases with the increase in relative density, and increases with the increase of the ratio of bond strength to fiber strength.

## Figures and Tables

**Figure 1 materials-17-02110-f001:**
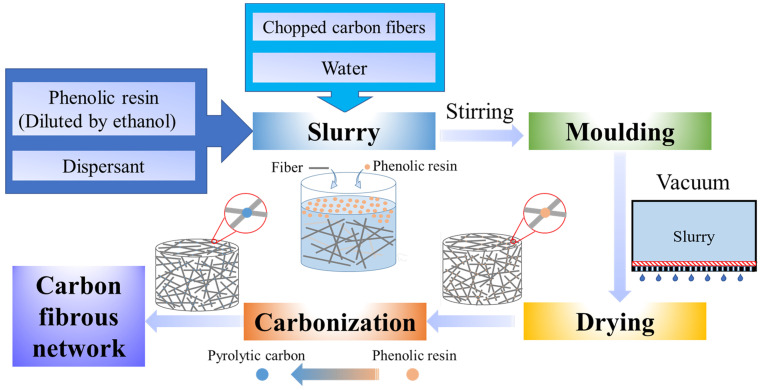
Fabrication process of carbon fibrous network materials.

**Figure 2 materials-17-02110-f002:**
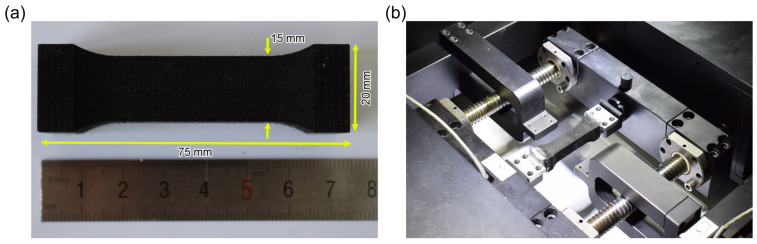
(**a**) Shape of fibrous network materials and (**b**) uniaxial tensile testing equipment.

**Figure 3 materials-17-02110-f003:**
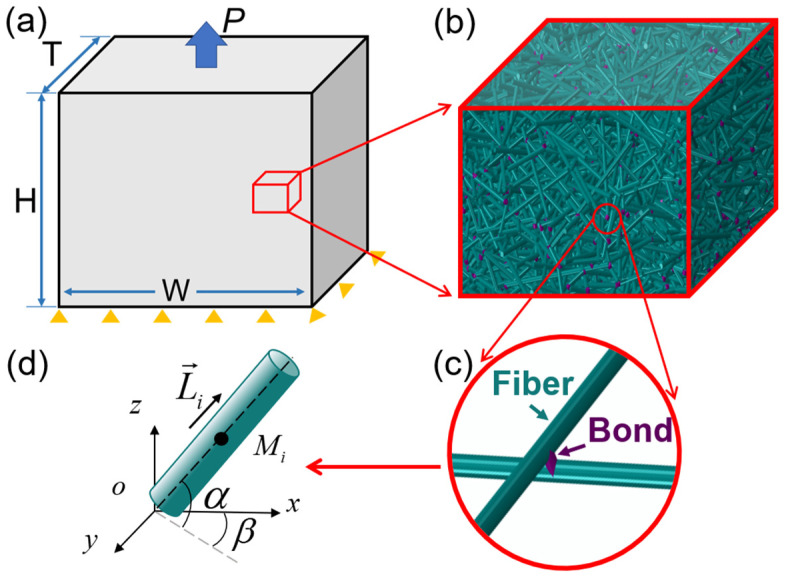
Numerical model of the fibrous network: (**a**) uniaxial tension of the RVE of 3D network model with size of *W* × *H* × *T*; (**b**) a close view of the fibrous network structure (**c**) with fibers colored in cyan and the bonds in purple; (**d**) a random fiber in the 3D space.

**Figure 4 materials-17-02110-f004:**
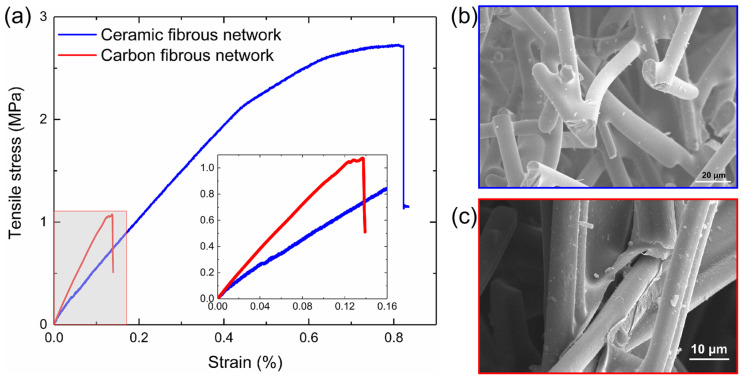
(**a**) Tensile stress–strain curves of the carbon fibrous network and the ceramic fibrous network. The SEM observation of the fracture surface of the specimens: (**b**) fiber damage domain in ceramic network and (**c**) bond damage domain in carbon network.

**Figure 5 materials-17-02110-f005:**
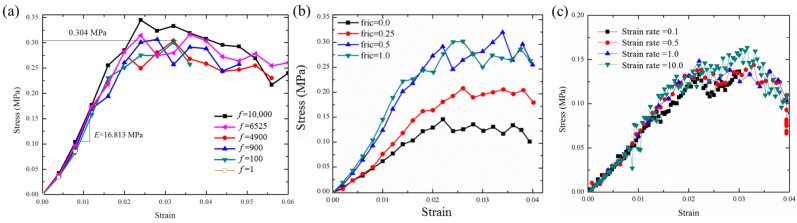
Discussion influence of (**a**) mass scale, (**b**) friction coefficient, (**c**) strain ratio on the stress–strain curve of 3D numerical model with 94% porosity in Explicit/CAE.

**Figure 6 materials-17-02110-f006:**
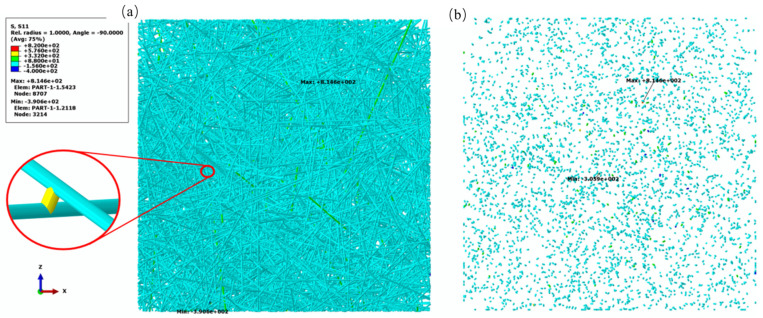
Principal stress on the (**a**) fiber and (**b**) bond in the fibrous network at a strain of 0.1%.

**Figure 7 materials-17-02110-f007:**
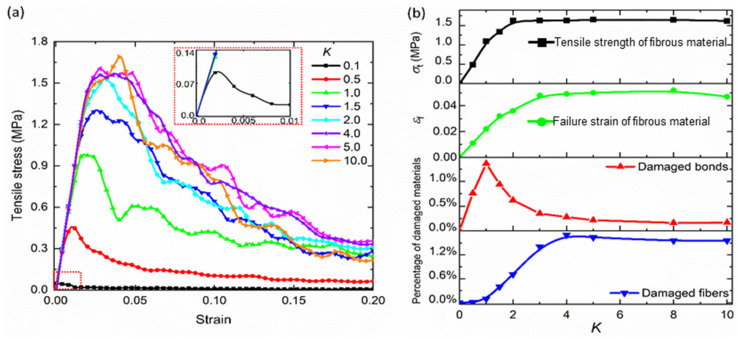
(**a**) Tensile stress–strain curves of numerical models with the *K* ranging from 0.1 to 10; (**b**) the tensile strength, failure strain of fibrous network, and percentage of damaged bonds and fibers in the network are obtained as functions of *K*.

**Figure 8 materials-17-02110-f008:**
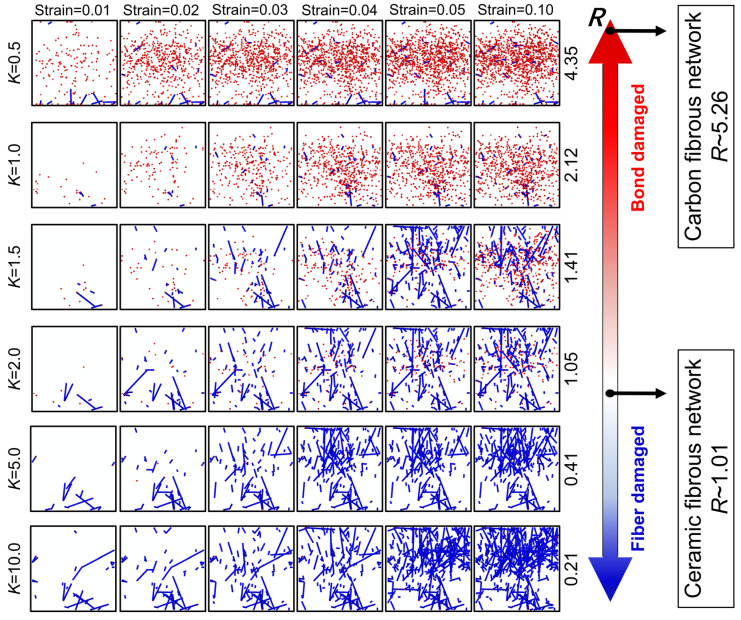
Damage distribution of the fibrous network with different *K*. The value *R* was marked to measure the failure mode of the network quantitatively. (The damaged bond elements are colored in red and the damaged fiber elements in blue).

**Figure 9 materials-17-02110-f009:**
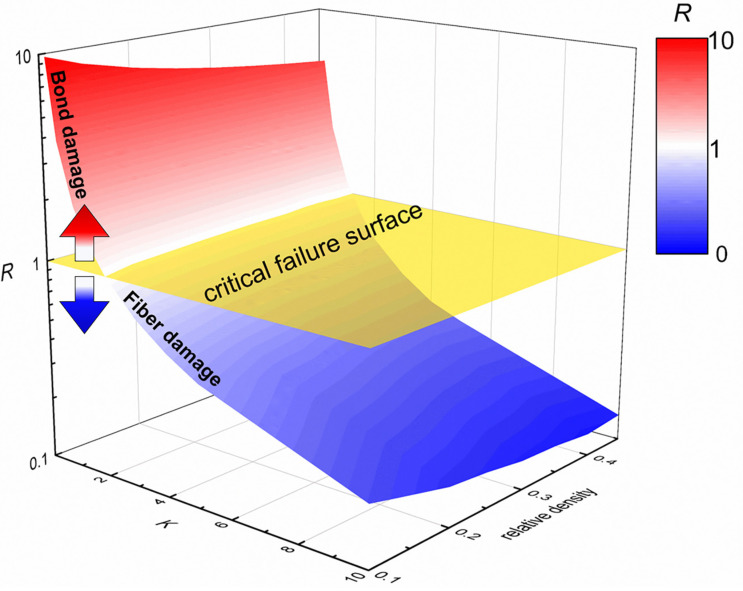
*R*-surface: the relationship among failure mode transition of network, relative density and variable *K*.

**Table 1 materials-17-02110-t001:** Material properties of the carbon fibrous network and ceramic fibrous network.

	Silica Fiber	Mullite Fiber	Carbon Fiber	B_2_O_3_ Bond	Pyrolytic Carbon Bond
Elastic modulus	78 GPa	193 GPa	230 GPa	27 GPa	4.65 GPa
Strength	1.7 GPa	2.0 GPa	3.5 GPa	2.0 GPa	0.2 GPa

**Table 2 materials-17-02110-t002:** Geometric parameters and material properties of the fibrous network.

Geometrical Parameters		Mechanical Properties	
Relative density (*ρ*)	0.08~0.45	The elastic modulus of the fibers (*E*_f_)	100 GPa
Fiber diameter (*D*)	9.0 μm	The elastic modulus of the bonds (*E*_b_)	100 GPa
Bond diameter	9.0 μm	The strength of the fibers (*σ*_f_)	1.0 GPa
Fiber length (*L*)	800 μm	The strength of the bonds (*σ*_b_)	0.1~10 GPa
Bond length	0~9.0 μm	Poisson ratio of the fibers and the bonds	0.26
*W* × *T* × *H*	1.5 *L* × 1.0 *L* × 1.5 *L*	The density of the fibers and the bonds	100 g/cm^3^

## Data Availability

Data are contained within the article.

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
