# Peer review of "Microstructural Evolution and Failure in Fibrous Network Materials: Failure Mode Transition from the Competition between Bond and Fiber"

_materials, 2024, doi:10.3390/ma17092110_

Round 1
Reviewer 1 Report
Comments and Suggestions for Authors
The analysis of failure mode transitions represents a long tail of studies in the framework of Fracture Mechanics of quasi-brittle solids and structures.
In particular, in the last decades many efforts have been devoted to the analysis of failure mode transitions in fibre-reinforced brittle-matrix composites: from fibre slippage to fibre rupture failure, from ductile to brittle failure, from softening to snap-back (cusp-catastrophe) behaviour.
The Authors are strongly encouraged to enhance their knowledge by deepen their study in this field. The Authors can refer, among other works, to: (1) A. Carpinteri, F. Accornero, “The Bridged Crack model with multiple fibres: Local instabilities, scale effects, plastic shake-down, and hysteresis”, Theoretical and Applied Fracture Mechanics, 2019, 104:102351; (2) F. Accornero, A. Rubino, A. Carpinteri, “Post-cracking regimes in the flexural behaviour of fibre-reinforced concrete beams”, International Journal of Solids and Structures, 2022, 248:111637; F. Accornero, A. Rubino, A. Carpinteri, “Ductile-to-brittle transition in fibre-reinforced concrete beams: Scale and fibre volume fraction effects”, Material Design & Processing Communications, 2020, 2(6):e127.
Concerning the numerical analysis, the Authors could better describe if their approach could take into account the main feature of a fracture mechanics problem, such as the one described in this manuscript, that is the energy dissipation over a surface (and not in a volume!). Is the proposed FEM approach related to a so-called "Bridged Crack" problem, in which the fibres are considered individually as bridging the two crack faces, or to a "Cohesive Crack" problem, in which the crack is interpreted as a localized zone of damage, across which a cohesive stress is transferred, representing the process of progressive damage (microcracking) and subsequent bridging due to fibres?
Finally, in Figure 6 a clear ductile-to-brittle transition is reported by the Authors. Do this transition be affected also by the specimen size? Is there a parameter that can synthetically describe a comprehenesive scale effect for fibre-reinforced composites? The Authors could refer to the Brittleness Number (please, see (4) A. Carpinteri, F. Accornero, A. Rubino, “Scale effects in the post-cracking behaviour of fibre-reinforced concrete beams”, International Journal of Fracture, 2023, 240(1):1-16, in addition to (1) and (2)).
Comments on the Quality of English LanguageModerate English editing in the whole manuscript is required.
Reviewer 2 Report
Comments and Suggestions for Authors
Review for “Materials MDPI”
Manuscript ID: materials-2861149
Title: Failure Mode Transition in Fibrous Network Materials from the Competition Between Bond and Fiber
In the present manuscript, the authors experimentally and numerically analyze the failure behavior of fibrous network materials and propose a simple failure criterion to identify if the material failure is due to the fiber or bond rupture. The authors use a 3D FE model, based on numerical approaches available in the technical literature thus not reporting aspects of novelty from a computational point of view. However, the topic is worthy of investigation and gathers numerical results interesting from a viewpoint of engineering design application and it presents a valuable contribution, possibly stimulating further research.
In my opinion, a Major revision is required before the publication on “Materials”, and some suggestions are proposed. To this end, the authors are encouraged to prepare a revised version in which the following indications should be considered while finalizing their paper:
1) The title 'Failure Mode Transition in Fibrous Network Materials from the Competition Between Bond and Fiber' might not fully encapsulate the breadth of your study. While it highlights the competition between bond and fiber in failure mode transition, your research delves deeper into the collapse behavior of fibrous network materials. Through a combination of experimental and numerical investigations, you explore various aspects of material collapse beyond the singular focus implied by the title. As such, a title revision may be warranted to better reflect the comprehensive nature of our findings.
2) I am intrigued by your proposed collapse criterion and its applicability to materials featuring diverse fiber types, such as silica fiber and mullite fiber. Could you please elaborate on how your criterion, explained by Equation (1) accounts for these variations in fiber composition and properties? Understanding how the criterion accommodates different fiber types will provide valuable insights into its robustness and versatility across a range of fibrous network materials.
3) In the introduction section, the author explains the different numerical models employed to simulate the mechanical behavior of brittle materials However, a small introduction on the nonlinear numerical models used in the literature to simulate the damage in brittle and quasi-brittle materials, seems to be missing. I suggest the author to discuss also the recent works on the fracture approaches (cohesive models and damage models) to simulate damage, including cracking behavior, which could affect the fibrous materials during the loading history. For example, the bibliographic context in such a Section of the paper could be properly enlarged improving the literature review of the recent numerical analyses. To this end, I recommend to discuss about these works in which the nonlinear models for damage propagation are proposed:
a. The Reinforcing Effect of Nano Modified Epoxy Resin on the Failure Behavior of FRP Plated RC Structures. Buildings 2023, 13, 1139. DOI: https://doi.org/10.3390/buildings13051139,
b. Uncertainty of the smeared crack model applied to RC beams, Engineering Fracture Mechanics 233 (2020) 107088, DOI: https://doi.org/10.1016/j.engfracmech.2020.107088,
c. Dynamic mode I and mode II crack propagation in fiber reinforced composites Mechanics of Advanced Materials and Structures 16(6), pp. 442-455, https://doi.org/10.1080/15376490902781183.
4) In the Section 2.2.1. Fibrous Network Model, at line 137, the Euler angles are introduced which are suitable illustrated in Figure 3d. Please report the correct Figure in the text. A similar comment should be taken into account for Line 140, where the bonding distance is clearly shown in Figure 3c and not in Figure 3d. Please correct them.
5) Line 167. The authors state that the details of quasi-static simulations are explained in the Supplement materials. I think some computational details should be inserted in the text, to better understand the numerical procedure employed for the simulations of the fibrous network material failure.
6) In Table 1 the authors report the material properties of the carbon and ceramic fibers. Could you please clarify the source of the material properties listed in the table? Are they obtained from experimental tests conducted by the authors in Section 2.1, or are they provided by the suppliers of the materials.
7) The parameter K, introduced at line 221 should be better explain adding an equation of the strength ratio.

1) Some typos are detected, please correct them:
a. Line 10, the word “here” should have the initial letter in capital letters;
b. Line 94, the word Dispersed in useless. Please remove it;
c. Line 96, remove the article “the” between the word “after” and “sufficiently”;
d. Line 104, the initial letter of the sentence between brackets should be in lowercase
e. Line 229, the initial letter of the word “with” should be in lowercase
f. And more…..
Reviewer 3 Report
Comments and Suggestions for Authors
The study raises the very important question of how to predict failures in the freely distributed multi-component system. The solution presented by the authors can be better understood if some points are clarified:
1. Line 137 - Euler angles - check the typing.
2. Please check the abbreviations in tables and figures to be sure there is no confusion (MPa and GPa).
3. Will be nice seeing a couple of words about considering of shear stress influence, regarding, for example sources 21 and 23.
4. What is meaing of bond lenght = 0. Do the fibers interact one another in this case?
Round 2
Reviewer 1 Report
Comments and Suggestions for Authors
Please, provide a sufficient scientific background related to Failure Mode Transition studies in the field of brittle and quasi-brittle materials. The research presented in this manuscript is only the last of a long tail of studies on ductile-to-brittle transitions and scale-transitional problems, all driven by the competition between different material properties, other than the size-scale.
Author Response
The literature recommended by the reviewers is mainly about fiber-reinforced concrete composite materials, in which fibers play a bridging and toughening role at material cracks. The fiber size is in the micrometer scale, and the crack size is in the millimeter scale, so there is a size effect. However, this article studies the fiber network structure, where both fibers and bonding materials belong to the same micrometer scale size. Material failure is also caused by the failure of fibers and adhesive materials, and there are no cracks or scale effects of component materials.
Reviewer 2 Report
Comments and Suggestions for Authors
I have thoroughly reviewed the manuscript and believe it is now suitable for publication. However, I would like to bring to your attention that the authors' responses to some of my queries, particularly Questions 3 and 7 (Parameter K), were not entirely convincing. I feel that the document could benefit from considering the suggestion I provided for the above-mentioned questions.
Author Response
This article cites some of the latest literature on fracture research of fiber-reinforced composite materials, such as
Huang Y, Qiu Y, Wei Y. Composite interlaminar fracture toughness imparted by electrospun PPO veils and interleaf particles: A mechanistical comparison[J]. Composite Structures, 2023,312:116865.
Zhang H, Huang Y, Xu S, et al. An explicit methodology of random fibre modelling for FRC fracture using non-conforming meshes and cohesive interface elements[J]. Composite Structures, 2023,310:116762.

Reviewer 3 Report
Comments and Suggestions for Authors
Authors provided comprehensive responses to all reviewer questions
